# Is Revision Arthroscopic Bankart Repair a Viable Option? A Systematic Review of Recurrent Instability following Bankart Repair

**DOI:** 10.3390/jcm13113067

**Published:** 2024-05-23

**Authors:** Alexander Baur, Jasraj Raghuwanshi, F. Winston Gwathmey

**Affiliations:** 1Liberty University College of Osteopathic Medicine, Lynchburg, VA 24502, USA; 2University of Virginia School of Medicine, Charlottesville, VA 22903, USA; 3Division of Sports Medicine, Department of Orthopaedic Surgery, University of Virginia Health System, Charlottesville, VA 22903, USA

**Keywords:** anterior shoulder dislocation, glenoid labrum repair, GBL, HSL, Latarjet technique

## Abstract

**Background/Objectives**: Recurrent shoulder instability following Bankart lesion repair often necessitates surgical revision. This systematic review aims to understand the failure rates of arthroscopic revision Bankart repair. **Methods**: Following the PRISMA guidelines and registered on PROSPERO, this systematic review examined twenty-five articles written between 2000 and 2024. Two independent reviewers assessed eligibility across three databases, focusing on recurrent instability as the primary endpoint, while also noting functional measures, adverse events, revision operations, and return-to-sport rates when available. **Results**: The key surgical techniques for recurrent instability post-Bankart repair were identified, with revision arthroscopic Bankart being the most common (685/1032). A comparative analysis revealed a significantly lower recurrence for open coracoid transfer compared to arthroscopic revision Bankart repair (9.67% vs. 17.14%; *p* < 0.001), while no significant difference was observed between remplissage plus Bankart repair and Bankart repair alone (23.75% vs. 17.14%; *p* = 0.24). The majority of studies did not include supracritical glenoid bone loss or engaging Hill–Sachs lesions, and neither subcritical nor non-engaging lesions significantly influenced recurrence rates (*p* = 0.85 and *p* = 0.80, respectively). **Conclusions**: Revision arthroscopic Bankart repair remains a viable option in the absence of bipolar bone loss; however, open coracoid transfer appears to have lower recurrence rates than arthroscopic Bankart repair, consistent with prior evidence. Further studies should define cutoffs and investigate the roles of critical glenoid bone loss and off-track Hill–Sachs lesions. Preoperative measurements of GBL on three-dimensional computed tomography and characterizing lesions based on glenoid track will help surgeons to choose ideal candidates for arthroscopic revision Bankart repair.

## 1. Introduction

Traumatic anterior shoulder instability is one of the most common conditions in orthopedic sports medicine, and arthroscopic repair is often the modality of choice for diagnosis and treatment. Recurrent instability is a major cause of revision presenting challenges due to distorted anatomy and difficulty in identifying anatomic lesions [1]. The optimal management approach for individuals who have experienced failure of their initial arthroscopic procedure remains a subject of controversy [2].

After an index arthroscopic primary Bankart repair (APBR), the options for surgical intervention for recurrent anterior shoulder instability most commonly include arthroscopic reversion Bankart repair (ARBR), open coracoid transfer (OCT) techniques such as the Bristow and Latarjet procedures, and arthroscopic Bankart repair with remplissage [3]. An international consensus study, known as the Delphi studies, identified different indications for these procedures [3]. In their study, they underscored the importance of two pivotal bone lesions in the decision-making process for instability. Specifically, the extent of glenoid bone loss (GBL) and the presence of a Hill–Sachs lesions (HSL) on the humerus significantly influence the decision-making process.

Bankart repair is indicated for primary or recurrent instability with a high risk for the failure of non-operative management. Specific imaging findings that suggest a successful Bankart repair are minimal GBL, on-track HSL, and magnetic resonance imaging (MRI) confirmation of labrum tear/Bankart lesion [3]. Remplissage is indicated for a large HSL either off-track or engaging at the time of arthroscopy [4]. Recently, the use of the glenoid track model for guiding surgical management has been questioned by Rashid et al. [5].

Bone block procedures have traditionally been reserved for more severe instability with significant bone loss. The Latarjet procedure is indicated for recurrent instability, failed prior surgery, contact athlete, critical GBL, and bipolar bone loss resulting in an off-track lesion [6]. Finally, free flap glenoid bone grafting such as the Eden–Hybinette procedure is indicated for critical bone loss and failed prior Latarjet procedure [7]. The precise threshold for critical GBL remains undefined. However, there is a consensus that, when greater than 15–20% of the glenoid circumference is compromised, it serves as a reliable indicator for considering OCT or bone grafting, particularly in severe cases [3].

Bankart versus Latarjet repair is a key discussion area, despite fundamental differences in technique. Bankart repair is arthroscopic, anchors avulsed labrum back into the anatomic position, and addresses soft tissue stability. The Latarjet procedure is usually open, uses a coracoid graft, and relies on a combination of an increased glenoid surface area and conjoint tendon sling effect to restore stability. The aims of this review are to (1) compare the instability recurrence rates for ARBR to other types of anterior stabilization, (2) determine the effect of bipolar bone loss on recurrent instability, and (3) provide a framework for managing recurrent anterior shoulder instability. The primary research question of this systematic review is to evaluate the effectiveness of ARBR after failed APBR. Additionally, we aim to draw comparisons with OCT techniques, the other commonly preferred modality for addressing recurrent instability.

## 2. Methods

PubMed and Google Scholar were selected as the study databases, given that these are both open source and ideal for study replication by other researchers. We utilized MeSH headings to query PubMed, which unites similar terms (such as “revision”, “repeat surgery”, and “reoperation”) under one controlled vocabulary (“reoperation”). Google Scholar was chosen for its comprehensiveness and ability to identify the greatest number of relevant citations. The search strategy was devised with intent to answer the question, “Does arthroscopic revision Bankart repair produce acceptable (<20%) recurrent instability rate?” The search strategy utilized for both databases was “Reoperation AND (Bankart repair OR Latarjet).” The date range analyzed was from 2000 through to the present time in 2024. Both paid and open-access articles were included in the search. Only studies published in English were included.

The study selection process engaged two independent reviewers (study authors A.B. and J.R.) who evaluated the titles and abstracts of the retrieved records for eligibility, and any disparities between reviewers were resolved through discourse or consultation with a third reviewer (senior author W.G.). From seventy-one articles initially identified, twenty-five articles were included in the systematic review after exclusion for duplicates, prior systematic reviews, and non-English-language articles (see Figure 1). The types of studies analyzed consisted of randomized controlled trials (RCTs) and non-randomized studies (cohort studies and case–control studies) with comparative designs. The inclusion criteria comprised the following: original research articles, revision surgery for failed arthroscopic primary Bankart repair, follow-up of six months or greater, defined instability (positive apprehension test, subluxation, and dislocation), and a reported recurrent instability rate. Revision surgical techniques included repeat arthroscopic and open Bankart repair, arthroscopic remplissage, and arthroscopic and OCT procedures (Bristow and Latarjet procedures). Comparative surgical techniques were considered as the comparators. The primary outcome of interest was recurrent shoulder instability. Functional outcome measures (e.g., range of motion, strength, and performance), patient-reported outcomes, adverse events, revision operations, and return-to-sport were analyzed when available. From the seventy-one studies identified, twenty-five were included in the systematic review (see Figure 1). Only two of the studies had follow-up periods of less than two years [8,9].

For data extraction, we utilized Microsoft Excel-365 2023 -Version 2401 (Microsoft Corporation; Redmond, WA, USA) tables specifically devised and piloted for this purpose. Two independent reviewers extracted the relevant details from the included studies, covering the study and participant characteristics, intervention specifics, outcomes, and adverse events. Discordances among the reviewers were resolved through discussion or consultation with a third reviewer.

The Risk of Bias in Nonrandomized Studies of Interventions (ROBINS-I) assessment tool (version 19, September 2016; The Cochrane Collaboration; London, UK) was utilized with a low risk of bias judgement in all categories, including confounding variables, selection bias, deviation from interventions, missing data, measurement of outcomes, and overall bias (see Table 1). Data analysis and synthesis encompassed the compilation of the extracted data into a master table, encompassing primary and secondary outcomes. An exploration of potential sources of heterogeneity was achieved through subgroup analyses and sensitivity analyses, which factored in considerations like surgical techniques, population characteristics (e.g., athlete type, competition level, age groups, and mechanism of injury), and surgical outcomes. The systematic review protocol was registered in the PROSPERO database (ID: 536764) to enhance transparency and avoid duplication. Full PRISMA 2020 checklist is included in Appendix A.

## 3. Results

Collectively, the studies analyzed in this review encompass a substantial patient population, with a total of 1032 patients across the included studies (See Table 2). Among these patients, 781 underwent arthroscopic procedures, while 251 patients underwent open surgeries. The most common procedure was the revision arthroscopic Bankart with 685 patients. There has been an increase in the use of remplissage concurrently with Bankart repair recently since 2020 (see Figure 2). There were 217 open Latarjet procedures.

Chi-squared testing showed a significantly higher recurrence rate (17% vs. 8%; *p* < 0.001) for revision arthroscopic Bankart repair compared to the open Latarjet procedure (see Figure 3). However, when remplissage was used to augment the Bankart repair (exclusively in studies which included patients with off-track HSLs), there was no significant difference in the recurrence rate compared to Latarjet (12% vs. 8%; *p* = 0.24).

The preoperative findings, recurrence rates, and conclusions for each study are reported in Table 1. There were many different findings reported across the papers, but the most common lesions identified preoperatively or intraoperatively were HSLs and Bankart lesions with reported GBL. All the studies that reported Bankart bone lesions had an average GBL of less than 20%.

We further analyzed the glenoid and humeral lesions that were reported. The most common cutoff for critical GBL was 20% (eight studies) with a mean cutoff of 23 ± 4.2%. Within the subcritical range, we subsequently categorized glenoid and humeral lesions as mild, moderate, and severe, using percentages for GBL and the Calandra classification for HSL. After categorization, we performed a one-way ANOVA test to analyze the effects of bone loss on recurrence rates. Neither the presence of a non-engaging HSL (*p* = 0.80) nor the severity of subcritical GBL (*p* = 0.85) had a statistically significant impact on the recurrence rate.

Regarding recurrence rates, it is important to note that persistent instability is one of the major risks associated with surgery. Among the included studies, only one study did not report on the rate of recurrence [18]. In this systematic review, the rate of recurrence ranged from 4.5% to 44%. The study by Boileau et al. reported the lowest recurrence rate of 4.5% (1/22) [26]. On the other hand, Elamo et al., Su et al., and Slaven et al. al reported recurrence rates above 40% [9,10,16]. These findings highlight the significant variability in recurrence rates observed across different patient populations and surgical techniques.

There was considerable variability in recurrence rates over time (see Figure 2). In the last 5 years, there have been four studies on arthroscopic Bankart revisions that reported recurrence rates above 20%. When analyzing the conclusions, the majority of studies concluded that arthroscopic Bankart revision surgery was satisfactory and an appropriate treatment option for patients with recurrent shoulder instability. However, there were two studies by Slaven et al. and Elamo et al. that concluded that revision Bankart procedures are not a viable option for recurrent instability due to the high recurrence rates [9,10]. In another study by O’Neill, they concluded that both Bankart procedures and Latarjet procedures have poor outcomes [15].

Four studies investigated recurrent instability between ARBR and OCT techniques. The mean recurrence rate was higher in ARBR (17.14%) across all studies compared to the mean recurrence rate in those with OCT cohorts (9.67%). Arthroscopic OCT procedures had an intermediate recurrence (12.45%). The pooled odds ratio for a higher instability recurrence in ARBR compared to OCT was 1.28 (95% CI: −1.4–3.96). One study was excluded from the odds ratio analysis due to being an outlier (see Figure 4). Elamo et al. (2020), which had an OCT recurrence rate of 0% and an ARBR recurrence rate of 43.3%, yielded an odds ratio of instability recurrence of 27.53 [9]. The Mantel–Haenszel correction method was utilized for analysis [33]. With the inclusion of this study, the pooled odds ratio of recurrent instability with ARBR rose to 8.75 (95% CI: 6.13–11.37).

Functional outcomes also offer important insights into the benefits of revision surgery for patients. The values consistently demonstrated improvements in the reported outcome measures following surgery. These findings suggest that arthroscopic Bankart repair can lead to positive functional outcomes, including improved shoulder stability and range of motion. Specifically, 13 studies reported on the patients returning to their sports. The rate of return to sport ranged from 50% to 91%. Four studies reported a return to sport rate exceeding 80%, indicating a favorable outcome [11,12,22,25]. Another approach to evaluating the impact on athletes is by assessing their strength, endurance, and range of motion. Bartl et al. conducted a study in which they observed significant improvements (*p*-value < 0.0001) in each of these categories following the revision compared to pre-revision measurements [22].

## 4. Discussion

We deliberately selected the year 2000 as the start of our systematic review, because it represented a major paradigm shift in the understanding of glenohumeral stability. Surgeons began to recognize the glenoid’s role via the concavity-compression mechanism and the biomechanical and therapeutic implications of critical GBL [34,35]. The treatment algorithm at that time was APBR as the index procedure and revision open Bankart repair for subcritical GBL. However, when the concept of critical GBL was introduced, surgeons began to consider OCT for supracritical GBL, leading to the resurgence of the Latarjet and Bristow procedures [34,36]. At this time, critical GBL was defined qualitatively as the “inverted pear”, although a cutoff of 25–30% correlated with this [34]. On the humeral side, the engaging HSL was defined in the pathogenesis of recurrent instability, and dynamic arthroscopic exam under anesthesia was believed to be the gold standard for determining engagement [34,37]. In the period immediately following this new understanding, surgeons began to quantify the recurrent instability rates of ARBR and how it compared to alternative revision options. Kim et al. (2002), Sisto et al. (2007), Creighton et al. (2007), and Neri et al. (2007) all discovered that ARBR has a less than 25% recurrence rate with good-to-excellent PROs [29,30,31,32]. By this time, they had already begun to exclude patients with supracritical GBL, which they defined as 25–30%. They also mostly excluded patients with engaging HSLs.

In 2007, the concept of the glenoid track was proposed, which underlies the biomechanics of engaging versus non-engaging HSLs [38]. The track was determined to be 83% of glenoid width, which is the region where the humeral head contacts the glenoid in 90° abduction and 0–135° external rotation [34,38]. This would still require time for studies to incorporate the track concept into the lexicon, with most studies preferring to utilize the prior nomenclature of “engaging” or “non-engaging” from dynamic arthroscopic exams. This period of studies illustrated even lower recurrence rates for ARBR, with most studies having a recurrence of less than 15%. These studies, Franceschi et al. (2008), Barnes et al. (2009), Boileau et al. (2009), Cho et al. (2009), Krueger et al. (2011), Ryu and Ryu (2011), Bartl et al. (2011), and Arce et al. (2012), began to measure HSL engagement, with most studies electing to exclude patients with engaging lesions [21,22,23,24,25,26,27,28]. Interestingly, the only two studies at this time that included engaging HSLs (Cho et al., 2009 and Ryu and Ryu, 2011) reported that all patients with recurrent instability also had engaging lesions. None of the patients in these cohorts received remplissage [23,25]. Ryu and Ryu (2011) also reported an abnormally high recurrence rate of 27%, perhaps confounded by the presence of engagement [23].

At approximately the same time, surgeons began to recognize the limitations of the dynamic arthroscopic exam. One cannot perform a robust exam after a Bankart repair for risk of jeopardizing the repair. Additionally, the conditions under anesthesia with the capsuloligamentous complex distended by arthroscopic fluid did not recapitulate physiology. From this arose a new need for predicting recurrent instability by preoperative risk factors [38]. Early attempts to achieve this focused on the presence of a large GBL and HSLs on plain radiographs via the instability severity index score (ISIS) score; however, this was subsequently shown to have a poor correlation with recurrent instability [39,40,41]. Giacomo et al. (2014) realized that the probability of engagement could be predicted by superimposing the predicted glenoid track on the HSL [37]. This defined a new area of studies which sought to characterize lesions as “off-track” rather than “engaging”, and while engagement was inconsistently reproduced under dynamic arthroscopic exam, the off-track lesion could be characterized on preoperative imaging. Notably, the amount of GBL also affects the glenoid track by reducing the contact area for the glenoid, and studies began to measure bipolar bone loss, with the percentage of GBL on the glenoid side and superimposing the glenoid track on the HSL on the humeral side.

From the late 2000s to early 2010s, there was a renaissance of GBL quantification methods, beginning initially with arthroscopic estimates, transitioning to the sagittal oblique projection on two-dimensional computed tomography (CT), and finally arriving at the gold standard of three-dimensional reconstructions of CT [42]. Although initially proposed in 2005, 3D CT was not the measurement tool of choice until approximately 2014 [42,43,44]. As cohorts in earlier studies underwent less accurate two-dimensional imaging, their GBL may have been underestimated, leading to improper indication for ARBR rather than OCT. All of the studies in this earlier period with a >25% recurrent instability utilized these less accurate measurement techniques: Kim et al. (2002) utilized arthroscopic estimation (critical GBL cutoff: 30%) and reported 21.7% recurrence, Ryu and Ryu (2011) used 2D CT (critical GBL cutoff: 20%) and reported 27% recurrence, and Neri et al. (2007) measured with a combination of MRI and plain radiograph (critical GBL cutoff: 30%) and reported 25% recurrence [23,29,32]. By 2016, most of the reviewed studies were utilizing 3D CT to quantify GBL, and these studies demonstrated a <20% instability recurrence, with 18% in Shin et al. (2016) and 12% in Buckup et al. (2018) [18,19]. Despite this, even with 3D CT, there are multiple measurement techniques with variable accuracy depending on the bony defect size and location, which contribute to our study heterogeneity [45].

After the improved ability to characterize osseous lesions on imaging, surgeons began to make direct head-to-head comparisons between ARBR versus OCT. Mahure et a. (2018) demonstrated that ARBR had a 12.4% recurrence rate compared to 5.1% in revision open stabilization, although this study grouped open Bankart repair and OCT techniques together for the analysis [17]. Elamo et al. (2020) found 43.3% recurrent instability in ARBR compared with zero cases of recurrence with the open Latarjet, which was the greatest difference across all studies and among the highest recurrence rate for ARBR. However, this study did not report the critical GBL cutoff for triaging to Latarjet [9]. Using a critical GBL cutoff on 20%, O’Neill (2020) found a concordant albeit weaker trend in the recurrent instability rates for ARBR versus open Latarjet (38% versus 29%) [15]. Calvo et al. (2021) was an interesting exception, in which ARBR had a slightly lower recurrence rate (11.8%) compared to the Latarjet (17.9%) [13]. The difference is that this study utilized arthroscopic Latarjet, which has been shown to have higher recurrence rates than open Latarjet, and the critical GBL cutoff was the lowest across all reviewed studies at 15% [11,13]. Slaven (2023) was another study that showed a very high recurrence rate of 44% after ARBR; however, this study may not be generalizable to the general population, as the cohort was a young, high-activity military population. Since recurrent instability is associated with contact sports and high-level athletics, military activity may be another risk factor [46].

Most recently, a small number of other systematic reviews have also been published analyzing ARBR outcomes. Yon et al. (2020) (14 studies) and Zhang et al. (2022) (19 studies) both reported a mean recurrence rate of 15.3%, which is very similar to our measurement of 17.14% [47,48]. Hong (2023) found a large range of recurrent instability from 6.1–46.8% across 11 studies, excluding studies which included a >20% bone loss in the cohort; however, a mean recurrence rate was not reported [2]. Lho et al. (2023) was the only other systematic review, to our knowledge, which reported mean recurrence rates for both ARBR (14.4%) and OCT (3.5%), finding a concordant trend with our study of decreased recurrence in OCT [6].

Furthermore, it is crucial to acknowledge the limitations of the studies included in our systematic review. Firstly, the absence of randomized controlled trials (RCTs), which are widely regarded as the gold standard for assessing treatment efficacy, is a notable limitation. Without RCTs, there is an increased risk of selection bias and confounding variables that may impact the observed outcomes. Many studies utilize small sample sizes, which may limit statistical power and increase the risk of Type II errors, thereby affecting the reliability of the findings. Other limitations include the inconsistent reporting of critical variables such as bone lesions, differing definitions of instability, and variations in follow-up duration, which can impact the accuracy and completeness of the data. These limitations collectively underscore the need for future research to address these gaps and provide more robust evidence to guide clinical decision making in managing failed primary stabilization.

Heterogeneity among the studies, including variations in patient demographics, surgical techniques, and outcome measures, also complicates direct comparisons and the generalizability of results. The presence of other concomitant injuries is a potential confounder and contributes to study heterogeneity. The studies had variable exclusion criteria regarding capsuloligamentous laxity, which has also been shown to influence recurrent instability; however, Bankart himself noted that the main pathology lies in the prevention of the anterior translation of the humeral head rather than the capsule [46,49]. Other associated conditions, such as superior labrum anteroposterior tears or rotator cuff tears, do not significantly influence recurrent instability, provided that they are addressed surgically at the time of ARBR [50]. There are also various technical considerations of ARBR which may influence recurrent instability, such as the number of sutures, non-anatomic repair (fixation proximal or medial to the glenoid margin), lack of anchors below the four o’clock position, or addressing concomitant pathology such as humeral avulsion of the glenohumeral ligaments; however, studies do not routinely report these intraoperative findings [46]. Various techniques have been proposed to augment ARBR, such as double-row repair with fixation to both the glenoid rim and neck, although biomechanical studies indicate that critical GBL will lead to recurrent instability, despite augmentation [46,51].

## 5. Conclusions

Overall, the last twenty-five years have led to a greater understanding of the role of bone loss and the glenoid track in recurrent instability. The osseous anatomy is a much larger driver than previously appreciated, and glenohumeral instability is a multifaceted problem beyond capsulolabral injury alone. OCT techniques that are over seventy years old, including the Latarjet procedure, have made a resurgence to address bone loss and provide an anterior buttress against humeral head subluxation. The advent of 3D CT has also improved our characterization of osseous lesions and ability to triage patients toward ARBR versus OCT. While we discovered that the overall recurrence rate (17.14%) for ARBR was within the acceptable range (<20%) for our original research question, arthroscopic (12.45%) and open (9.67%) OCT techniques (Bristow or Latarjet) had lower recurrence rates compared to ARBR. Consistent with prior studies, the evidence continues to suggest that OCT is the definitive treatment after a primary Bankart repair necessitating surgical revision, especially in cases of supracritical (>20%) bone loss.

## 6. Future Directions

Further guidance in selecting the most effective technique for arthroscopic revision stabilization could be provided through additional cohort studies. Additionally, the absence of randomized controlled trials for revision shoulder arthroscopy is a significant limitation, and conducting such trials would be invaluable in this context. We recommend further research on the impact of bone lesions on stabilization failure, given the current lack of a significant link. Future studies should also specifically compare OCT versus ARBR plus remplissage to understand how OCT techniques address engaging HSLs with the hope of finding more clear patient selection criteria for each procedure.

## Figures and Tables

**Figure 1 jcm-13-03067-f001:**
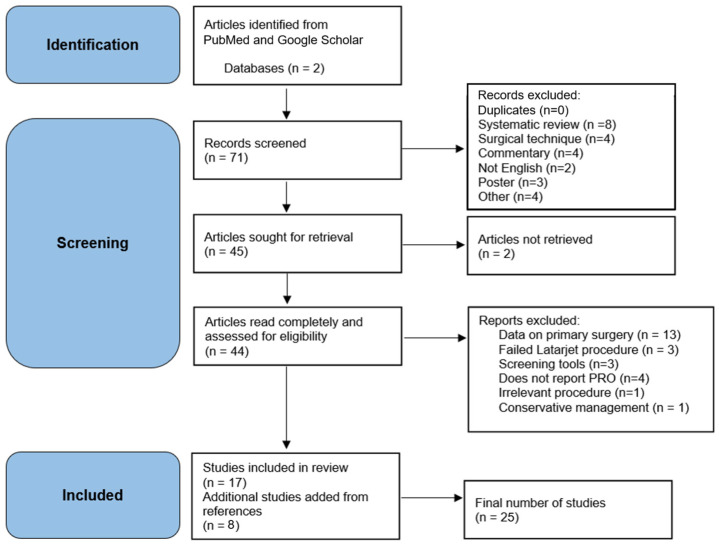
PRISMA flow diagram. Legend: We utilized the search inclusion and exclusion criteria per PRISMA guidelines as above. Seventy-one peer-reviewed articles were screened to remove prior systematic review articles, surgical technique articles, or original research articles that did not present recurrent instability data, leading to twenty-five articles included in systematic review.

**Figure 2 jcm-13-03067-f002:**
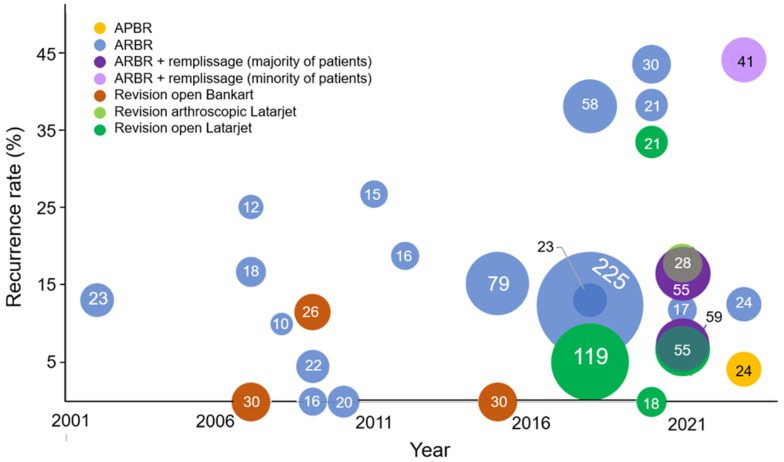
Recurrence rates for revision stabilization procedures over time. Legend: This figure illustrates the temporal trends of reported recurrence rates (expressed as a percentage) in revision surgery following a Bankart repair. The size of each bubble corresponds to the sample size of the respective studies included in the analysis. Arthroscopic primary Bankart repair, APBR; arthroscopic revision Bankart repair, ARBR.

**Figure 3 jcm-13-03067-f003:**
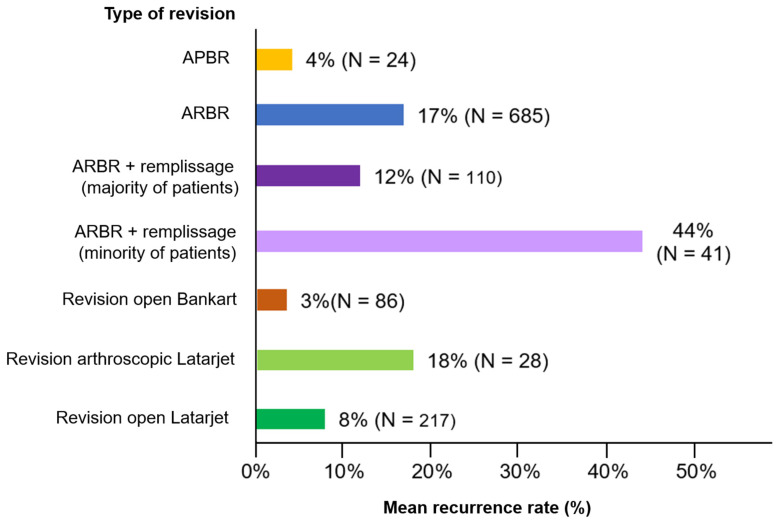
Comparison of mean recurrence rates by procedure type. Legend: The mean instability recurrence rates aggregated across the reviewed studies are plotted according to type of revision procedure and the sample size. Arthroscopic primary Bankart repair, APBR; arthroscopic revision Bankart repair, ARBR.

**Figure 4 jcm-13-03067-f004:**
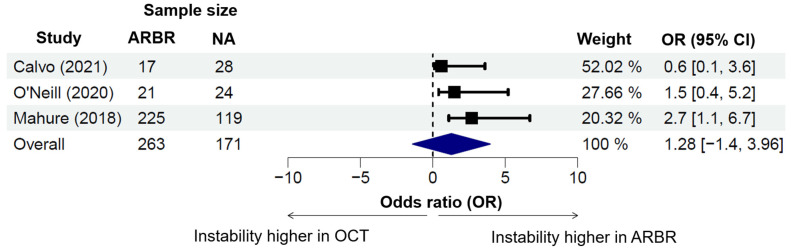
Recurrent instability between arthroscopic revision Bankart repair versus non-anatomic techniques. Legend: Forest plot comparing instability recurrence rates for arthroscopic revision Bankart repair (ARBR) to OCT techniques (OCT). The *y*-axis shows the odds ratio, with positive values representing higher instability with ARBR, and negative values representing higher instability with the Latarjet procedure [13,15,17].

**Table 1 jcm-13-03067-t001:** Appraisal using the Cochrane ROBINS-I Risk of Bias Assessment Tool.

Author (Year)	Confounding	Selection of Participants	Classification of Interventions	Deviations from Intended Interventions	Missing Data	Measurement of Outcomes	Selection of the Reported Result	Overall Bias	Notes
Lee (2023) [8]	+	+	+	+	+	+	+	Low-Risk	
**Slaven (2023)** [10]	**?**	**−**	**+**	**+**	**+**	**+**	**+**	**High-risk**	**Lack of matching; Military sample**
**Clowez (2021)** [11]	**−**	**+**	**−**	**+**	**+**	**+**	**+**	**High-risk**	**Variable surgery (Latarjet or Bristow)**
Sinha (2021) [12]	+	+	+	+	+	+	+	Low-Risk	
Calvo (2021) [13]	?	+	+	+	+	+	+	Uncertain	Lack of matching
Park (2021) [14]	+	+	+	+	+	+	+	Low-Risk	
O’Neill (2020) [15]	+	+	+	+	+	+	+	Low-Risk	
**Elamo (2020)** [9]	**−**	**+**	**+**	**+**	**+**	**+**	**+**	**High-risk**	**GBL cutoff and engaging HSL not reported**
**Su (2018)** [16]	**−**	**+**	**+**	**+**	**+**	**+**	**+**	**High-risk**	**Variable index surgery**
**Mahure (2018)** [17]	**−**	**+**	**?**	**+**	**+**	**+**	**+**	**High-risk**	**Variable surgery (ARBR, OCT, capsulorrhaphy)**
Buckup (2018) [18]	+	?	+	+	+	+	+	Uncertain	Male-only sample
Shin (2016) [19]	+	+	+	+	+	+	+	Low-Risk	
**Neviaser (2015)** [20]	**−**	**?**	**+**	**+**	**+**	**+**	**+**	**High-risk**	**Lack of matching; Variable index Surgery**
Arce (2012) [21]	?	+	+	+	+	+	+	Uncertain	Male-only sample
Bartl (2011) [22]	+	+	+	+	+	+	+	Low-Risk	
Ryu (2011) [23]	+	+	+	+	+	+	+	Low-Risk	
**Krueger (2011)** [24]	**−**	**?**	**+**	**+**	**+**	**+**	**+**	**High-risk**	**Lack of matching; Variable index surgery**
Cho (2009) [25]	+	+	+	+	+	+	+	Low-Risk	
Boileau (2009) [26]	+	+	+	+	+	+	+	Low-Risk	
Barnes (2009) [27]	+	?	+	+	+	+	+	Uncertain	Lack of matching
Franceschi (2008) [28]	+	+	+	+	+	+	+	Low-Risk	
Neri (2007) [29]	+	+	+	+	+	+	+	Low-Risk	
**Creighton (2007)** [30]	**−**	**+**	**+**	**+**	**+**	**+**	**+**	**High-risk**	**Variable Index Surgery**
Sisto (2007) [31]	+	+	+	+	+	+	+	Low-Risk	
Kim (2002) [32]	+	+	+	+	+	+	+	Low-Risk	

Legend: Low-risk (+); Uncertain (?); **High-risk (−)**. Bolded indicates study is high-risk of bias.

**Table 2 jcm-13-03067-t002:** Summary of Study Characteristics and Instability Recurrence Rates.

Author (Year)	N	Minimum Follow-Up (Months)	Experimental Group (N)	Control Group (N)	Recurrent Instability Definition	Experimental Recurrence (%)	Control Recurrence (%)	Critical GBL (%)	Critical GBL Treatment	Off-Track Hill-Sachs Lesion Treatment	Conclusions
Lee (2023) [8]	48	6	ARBR (24)	APBR (24)	A/S/D	12.5	4.2	20	Excluded	Included (remplissage)	ARBR has a nonsignificant (*p* = 0.06) but increased recurrence and decreased capsulolabral height compared to APBR
Slaven (2023) [10]	41	24	ARBR (41)	−	S/D	44	−	20	Excluded	Included	Recurrence rate is approximately 50% in a young (22.9 ± 4.3 yrs) military population
Clowez (2021) [11]	59	24	Arthroscopic OCT (34)	Open coracoid transfer (25)	S/D	7	0	−	−	Included	Arthroscopic has increased recurrence compared to OCT, and recurrence is related to Calandra grade III HSLs
Sinha (2021) [12]	42	24	ARBR with remplissage (42)	−	S/D	9.5	−	25	Excluded	Included	ARBR with remplissage is associated with recurrence <10% for patients with off-track HSLs
Calvo (2021) [13]	45	24	ARBR (17)	Arthroscopic Latarjet (28)	S/D	11.8	17.9	15	Included (Latarjet)	Included	ARBR and arthroscopic Latarjet have similar recurrence rates regardless of GBL; however, this study used a lower threshold than most others (15% versus 25%)
Park (2021) [14]	55	24	ARBR for capsular tear (10)	ARBR for labral retear (45)	D	40	10.2	25	Excluded	Included (all in labral retear group)	Capsular tears with healed labra are associated with increased recurrence compared to labral retears
O’Neill (2020) [15]	45	24	ARBR with remplissage (21)	Open Latarjet (24)	A/S/D	38	29	20	Included (Latarjet)	Included	ARBR with remplissage for off-track HSLs and open Latarjet for critical GBL have similar recurrence rates.
Elamo (2020) [9]	48	12	ARBR (30)	Open Latarjet (18)	S/D	43.3	0	−	−	−	ARBR has increased recurrence compared to open Latarjet; however, neither the cutoff nor the number of patients for critical GBL were defined.
Su (2018) [16]	92	24	ARBR (92)	−	S/D	42	−	20	Included (ARBR)	Included	Recurrence was associated with off-track lesions and capsulolabral insufficiency, and ARBR recurrence rate is <20% when excluding these factors
Mahure (2018) [17]	344	36	ARBR (225)	Revision open stabilization (119)	D	12.4	5.1	−	−	−	ARBR has increased recurrence compared to open restabilization; however, neither the cutoff nor the number of patients for critical GBL were defined.
Buckup (2018) [18]	47	24	ARBR (25)	Healthy controls (22)	D	12	−	20	Excluded		ARBR is associated with chronic atrophy of supraspinatus and infraspinatus.
Shin (2016) [19]	122	24	ARBR (89)	APBR (33)	A/S/D	18	3	25	Excluded	Excluded	ARBR has a significantly higher recurrence rate (*p* = 0.039) compared to APBR
Neviaser (2015) [20]	30	120	ARBR (30)	−	A/S/D	0	−	−	−	Included	ARBR has negligible recurrent instability with good-to-excellent PROs in the majority of patients
Arce (2012) [21]	16	24	ARBR (16)	−	S/D	18.8	−	25	Excluded	Excluded	ARBR has <20% recurrence and good-to-excellent PROs in the majority of patients
Bartl (2011) [22]	56	24	ARBR (56)	−	S/D	11	−	20	Excluded	Excluded	ARBR has <15% recurrence and good-to-excellent PROs in the majority of patients
Ryu (2011) [23]	15	18	ARBR (15)	−	S/D	27	−	20	Included (ARBR)	Included	ARBR has <30% recurrent instability, and recurrence is not related to critical GBL
Krueger (2011) [24]	40	24	ARBR (20)	APBR (20)	A/S/D	10	0	25	Excluded		ARBR has an increased risk of recurrent instability and poorer patient-reported outcomes compared to APBR
Cho (2009) [25]	26	24	ARBR (26)	−	A/S/D	11.5	−	20	Included (ARBR)	Included	ARBR has <15% recurrence and fair PROs in the majority of patients
Boileau (2009) [26]	22	24	ARBR after open index (22)	−	A/S/D	13.6	−	25	Excluded	Excluded	ARBR has <15% recurrence and good-to-excellent PROs in the majority of patients
Barnes (2009) [27]	18	24	ARBR (18)	−	D	5.6	−	−	−	−	ARBR has <10% recurrence and good-to-excellent PROs in the majority of patients
Franceschi (2008) [28]	10	46	ARBR (10)	−	D	10	−	30	Excluded	Excluded	ARBR has 10% recurrent instability and good-to-excellent PROs in the majority of patients
Neri (2007) [29]	12	24	ARBR (12)	−	S/D	25	−	30	Excluded	Excluded	ARBR has 25% recurrent instability and good-to-excellent PROs in the majority of patients
Creighton (2007) [30]	18	24	ARBR (18)	−	S/D	16.7	−	25	Excluded	Excluded	ARBR has <20% recurrent instability and good-to-excellent PROs in the majority of patients
Sisto (2007) [31]	30	24	ARBR (30)	−	A/S/D	0	−	−	−	Excluded	ARBR has negligible recurrent instability with good-to-excellent PROs in the majority of patients
Kim (2002) [32]	23	24	ARBR (23)	−	A/S/D	21.7	−	30	Excluded	Included	ARBR has <25% recurrent instability and good-to-excellent PROs in the majority of patients

Legend: Arthroscopic revision Bankart repair, ARBR; Arthroscopic primary Bankart repair, APBR; GBL, GBL; Apprehension, A; Subluxation, S; Dislocation, D; Patient-reported outcomes, PROs.

## Data Availability

Data extracted from included studies, data used for all analyses, analytic code, and any other materials used in the review can be obtained by contacting the corresponding author.

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
