# Peer review of "Is Revision Arthroscopic Bankart Repair a Viable Option? A Systematic Review of Recurrent Instability following Bankart Repair"

_jcm, 2024, doi:10.3390/jcm13113067_

Round 1

Reviewer 1 Report

Comments and Suggestions for Authors

I would like to thank the authors for their contributions to science. The study was prepared according to the PRISMA guidelines. However, there are some points in the article that need to be corrected.

1-Line 36-37.. "Recurrent instability is a major cause of revision presenting challenges due to distorted anatomy and difficulty in identifying anatomic lesions." 

No reference was given. The current reference below should be added. https://doi.org/10.3390/jcm12154910

2-line 61-62-63.. ".....15-30%...." 
No reference was given, an up-to-date reference should be added.

3-line 98...More detailed information about Excel should be given. e.g. Microsoft Excel. 2023. Microsoft Corporation.  
(It should be written under the journal criteria.)

4-line 153..Which writing style is correct?  p=0.80 or p=.85?

5-There is no paragraph about limitation in the study.

Author Response

I would like to thank the authors for their contributions to science. The study was prepared according to the PRISMA guidelines. However, there are some points in the article that need to be corrected.

1-Line 36-37.. "Recurrent instability is a major cause of revision presenting challenges due to distorted anatomy and difficulty in identifying anatomic lesions." 

No reference was given. The current reference below should be added. https://doi.org/10.3390/jcm12154910

Reference was added

2-line 61-62-63.. ".....15-30%...." 
No reference was given, an up-to-date reference should be added.

Added appropriate reference and reworded to be more clear.

“The precise threshold for critical glenoid bone loss remains undefined. However, there is a consensus that when greater than 15-20% of the glenoid circumference is compromised, it serves as a reliable indicator for considering coracoid transfer or bone grafting, particularly in severe cases[4].” 

3-line 98...More detailed information about Excel should be given. e.g. Microsoft Excel. 2023. Microsoft Corporation.  
(It should be written under the journal criteria.)

Added Microsoft excel version.

For data extraction, we utilized Excel-365 (2023) tables specifically devised and piloted for this purpose. Two independent reviewers extracted relevant details from included studies, covering study and participant characteristics, intervention specifics, outcomes, and adverse events.”

4-line 153..Which writing style is correct?  p=0.80 or p=.85?

Changed all p-value reporting to p=0.xx

5-There is no paragraph about limitation in the study.

Added paragraph on limitations

“Furthermore, it's crucial to acknowledge the limitations of the studies included in our systematic review. Firstly, the absence of randomized controlled trials (RCTs), which are widely regarded as the gold standard for assessing treatment efficacy, is a notable limitation. Without RCTs, there is an increased risk of selection bias and confounding variables that may impact the observed outcomes. Secondly, the heterogeneity among the studies, including variations in patient demographics, surgical techniques, and outcome measures, complicates direct comparisons and generalizability of results. Additionally, many studies suffer from small sample sizes, which may limit statistical power and increase the risk of Type II errors, thereby affecting the reliability of the findings. Other limitations include inconsistent reporting of critical variables such as bone lesions, differing definitions of instability, and variations in follow-up duration, which can impact the accuracy and completeness of the data. These limitations collectively underscore the need for future research to address these gaps and provide more robust evidence to guide clinical decision-making in managing failed primary stabilization.

Reviewer 2 Report

Comments and Suggestions for Authors

Thank you for the opportunity to review this paper Is Revision Arthroscopic Bankart Repair a Viable Option? A Systematic Review of Recurrent Instability Following Bankart Repair. Interesting study, given the subject matter. Congratulations on the choice of topic. I have a few suggestions for improvement to make the paper clearer and more interesting.

Abstract

In the abstract, it would be pertinent to include the period to which the systematic review refers in the methodology section and the actual number of articles included for analysis in the results. In the keywords, they shouldn't be repeated with the words that are already in the title, in order to increase the dissemination of the paper.

Methods: 

The inclusion criteria are unclear, and it is not clear between which years the studies were selected. Nor are other filters applied, such as freely available papers.It would be pertinent to explain why they only used two databases, and why these two databases. It is necessary to know whether the search expression was used for both databases, or whether there was a change in one of the databases, and what is the exact search expression. A very clear research question is lacking.

Where are the results of the Cochrane Risk of Bias tool? It would be pertinent to see that analysis.

It is important to know if the reviewers of the articles are the authors, and to identify the two reviewers and the third reviewer with the initials of their names. 

Results: 

In figure 1, the Google database is not identified as the database used; this should be reviewed. Were there no duplicate articles in the search? If so, you should post that information.

Figure 4 is not very clear, can you please add information to the figure for better clarity?

Discussion: 

Given the variability of the studies included and the variability of the results, I suggest that you organise the discussion better, based more on table 1.

Table 1 should have additional information to help outline and organise the discussion, and has unclear information.

Conclusions: 

The conclusion of a systematic review does not conclude on associations, that is not the aim of a systematic review. The conclusion does not seem to answer the objectives.

Comments on the Quality of English Language

 Minor editing of English language required

Author Response

  • Abstract
  • In the abstract, it would be pertinent to include the period to which the systematic review refers in the methodology section and the actual number of articles included for analysis in the results. 
  • Please see the updated change which reflects the dates and number of articles analyzed: “Following PRISMA guidelines and registered on PROSPERO, this systematic review examined twenty-four articles written between 2000 and 2024.”
  • In the keywords, they shouldn't be repeated with the words that are already in the title, in order to increase the dissemination of the paper.
  • Keywords have been updated so as not to overlap with the title. The updated keywords are: “Keywords (5): anterior shoulder dislocation, glenoid labrum repair, glenoid bone loss, Hill-Sachs lesion, Latarjet technique.”
  • Methods: 
  • The inclusion criteria are unclear, and it is not clear between which years the studies were selected.
  • Inclusion criteria is now explicitly stated: “All studies included a cohort that had failed a primary Bankart repair.” à “Inclusion criteria comprised the following: revision surgery for failed arthroscopic primary Bankart repair, follow-up of two years or greater, defined instability (positive apprehension test, subluxation, dislocation), and reported recurrent instability rate.”
  • The date range has now been explicitly stated: “The date range analyzed was 2000 through the present time in 2024.”
  • Nor are other filters applied, such as freely available papers.
  • The authors were able to access both paid and freely available articles as a result of institutional library access. We now specify that our analysis included these: “Both paid and open access articles were included in the search.”
  • It would be pertinent to explain why they only used two databases, and why these two databases. 
  • We added the following rationale for our selection of these two databases: “PubMed and Google Scholar were selected as the study databases, given that these are both open source and ideal for study replication by other researchers. We utilized MeSH headings to query PubMed, which unites similar terms (such as "revision", "repeat surgery", and "reoperation") under one controlled vocabulary ("reoperation").  Google Scholar was chosen for its comprehensiveness and ability to identify the greatest number of relevant citations.”
  • It is necessary to know whether the search expression was used for both databases, or whether there was a change in one of the databases, and what is the exact search expression. 
  • Regarding the search strategy used for each database, the following sentence was added: “The search strategy utilized for both databases was ‘Reoperation AND (Bankart repair OR Latarjet).’ “
  • A very clear research question is lacking.
  • The research question has been added in the description of the search strategy: “The search strategy was devised with intent to answer the question, ‘Does revision Bankart repair produce acceptable (<20%) recurrent instability rate?’ ”
  • Where are the results of the Cochrane Risk of Bias tool? It would be pertinent to see that analysis.
  • A new table was added to display the risk of bias results (now Table 1) with the following in-text reference: “The Risk of Bias in Nonrandomized Studies of Interventions (ROBINS-I) assessment tool (version 19, September 2016; The Cochrane Collaboration; London, England) was utilized with a low risk of bias judgement in all categories including confounding variables, selection bias, deviation from interventions, missing data, measurement of outcomes, and overall bias (see Table 1).”
  • It is important to know if the reviewers of the articles are the authors, and to identify the two reviewers and the third reviewer with the initials of their names. 
  • The article reviewers have now been specified as the study authors by their initials: “The study selection process engaged two independent reviewers (study authors A.B. and J.R.) who evaluated titles and abstracts of retrieved records for eligibility based on predefined criteria.” and “Any disparities between reviewers were resolved through discourse or consultation with a third reviewer (senior author W.G.).”
  • Results: 
  • In figure 1, the Google database is not identified as the database used; this should be reviewed. Were there no duplicate articles in the search? If so, you should post that information.
  • Please excuse us for this misprint. The correct databases are now reflected in the first box (Identification) of Figure 1.  Additionally, in the Screening phase, the absence of duplicate articles is denoted. 
  • Figure 4 is not very clear, can you please add information to the figure for better clarity?
  • The figure has been removed to avoid confusion with in-text reference and the figure order updated.
  • Discussion: 
  • Given the variability of the studies included and the variability of the results, I suggest that you organise the discussion better, based more on table 1.
  • Please find the new Discussion section, which has has been reorganized to align with the study order and results presented in Table 1.
  • Table 1 should have additional information to help outline and organise the discussion, and has unclear information.
  • Please find Table 1 now updated to clarify sample size, experimental and control groups, mean follow-up, recurrence rates, and conclusions.
  • Conclusions: 
  • The conclusion of a systematic review does not conclude on associations, that is not the aim of a systematic review. The conclusion does not seem to answer the objectives.
  • Please find the new Conclusion section which specifically addresses our research question, “Does arthroscopic revision Bankart repair produce acceptable (<20%) recurrent instability rate?”

Round 2

Reviewer 1 Report

Comments and Suggestions for Authors

Revisions have been made. It is appropriate to publish the work in its current form.

Author Response

Thank you for the helpful comments and suggestions.

Reviewer 2 Report

Comments and Suggestions for Authors

Thanks again for the opportunity to review the paper.

Congratulations on the changes made, it is much clearer.

I still don't understand table 1. I would suggest putting the risk of bias of each study in a qualitative way (Low risk, Uncertain and High risk), as it helps to interpret table 1 better. 

Author Response

I still don't understand table 1. I would suggest putting the risk of bias of each study in a qualitative way (Low risk, Uncertain and High risk), as it helps to interpret table 1 better. 

Changes have been made to table 1. Specifically, the overall bias column has been clarified. High-risk have also been bolded.